# Efficacy of virtual reality balance training on rehabilitation outcomes following anterior cruciate ligament reconstruction: A systematic review and meta-analysis

**Chao Du[1,2☯], Nei-Meng Gu[1,2☯], Tian-Ci Guo[1,2], Ai-Feng Liu[1,2]***

**1** Orthopedics Department, First Teaching Hospital of Tianjin University of Traditional Chinese Medicine, Tianjin, China, **2** National Clinical Research Center for Traditional Chinese Medicine and Acupuncture, Tianjin, China

☯ These authors contributed equally to this work.
* draifeng@163.com

## Abstract

### Objective

The objective of this systematic review and meta-analysis is to clarify the rehabilitation efficacy of virtual reality (VR) balance training after anterior cruciate ligament reconstruction (ACLR).

### Methods

This meta-analysis was registered in PROSPERO with the registration number CRD42024520383. The electronic databases PubMed, Web of Science, Cochrane Library, MEDLINE, Embase, China National Knowledge Infrastructure, Chinese Biomedical Literature, China Science and Technology Journal Database, and Wanfang Digital Periodical database were systematically searched to identify eligible studies from their inception up to January 2024. The investigated outcomes included International Knee Documentation Committee (IKDC) score, visual analogue scale (VAS), Holden grading, Extensor peak torque (EPT), Flexor peak torque (FPT), knee reaction time, knee reproduction angle difference. The pooled mean difference (MD) and 95% confidence intervals (CIs) were calculated using the random-effects model.

### Results

Six RCTs with a total of 464 patients after unilateral ACLR were included for 8–12 weeks of VR balance training intervention. Analysis of the results showed that compared with the conventional rehabilitation control group, the VR balance training group significantly improved the International Knee Documentation Committee (IKDC) score (MD = 3.88, 95%CI: 0.95~6.81), Holden grading (MD = 0.42, 95%CI: 0.33~0.51), Extensor peak torque (EPT) (MD = 12.03, 95%CI: 3.28~20.78)and Flexor peak torque (FPT) (MD = 14.57, 95%CI: 9.52~19.63) in postoperative ACLR patients, and significantly reduced knee reaction time

**Data Availability Statement:** All relevant data are within the paper and its Supporting information files.

**Funding:** The funding support from the National Natural Science Foundation of China (Grant no. 81873316) and the Tianjin Health Commission Jinmen medical talent project (Grant no. TJSJMYXYC-D2-028) for this work is gratefully acknowledged.

**Competing interests:** The authors have declared that no competing interests exist.

(MD = -0.30, 95%CI: -0.35~-0.25), knee angle reproduction difference at 30˚ (MD = -0.88, 95%CI: -1.16~-0.61), knee angle reproduction difference at 60˚ (MD = -0.80, 95%CI: -1.09~-0.50), and VAS score (MD = -0.52, 95%CI: -0.65~-0.39).

## Conclusion

Since many of the included results are based on low—or very—low—quality evidence, although the results show a certain trend, the conclusion has great uncertainty. In the rehabilitation training following ACLR and lower—limb balance training, the application of VR might be advantageous for the recovery of patients' knee joint function, lower—limb muscle strength, proprioception, and pain management. The level of immersion may influence the rehabilitation outcome. Because of the limitations in data quality and heterogeneity as well as the small sample size, the strength of the conclusions is weakened. These findings should be verified in further large-scale prospective studies.

## Introduction

Anterior cruciate ligament (ACL) injury is a common sports injury that can greatly influence an individual's knee function and overall quality of life. The ACL injuries are relatively prevalent among the younger population. Among those aged 16 to 39 years old, the prevalence is 85 per 100,000 people, especially for athletes engaged in sports that demand pivoting, cutting, or jumping motions [1]. The ACL is of crucial importance in maintaining the stability of the knee joint by controlling anterior—posterior and rotational motions. When the ACL is ruptured, it impairs the stability of the knee joint, resulting in instability, pain, swelling, and restricted mobility and function.

Anterior cruciate ligament reconstruction (ACLR) is the primary surgical intervention for restoring stability to the knee joint post-injury [2]. Although ACLR is a frequently—performed procedure, the proportion of patients who successfully return to their pre—injury functional state is only approximately 24% [3]. This highlights the significance of postoperative rehabilitation in maximizing outcomes and guaranteeing a successful resumption of activities. Typically, postoperative rehabilitation after ACLR involves a combination of interventions, including knee braces, progressive muscle exercises, weight—bearing exercises, knee mobility training, and knee proprioception training. These components are designed to enhance muscle strength, joint stability, range of motion, proprioception, and the overall functional recovery of the knee joint [4]. However, traditional postoperative rehabilitation programs are challenging for patients. The process is time—consuming, the exercises are repetitive and sometimes monotonous, and there are patient compliance issues. As a result, researchers and clinicians are exploring alternative rehabilitation modalities to improve outcomes and patient engagement and adherence to rehabilitation [5, 6].

One emerging technology is virtual reality (VR) therapy. It creates a simulated environment for individuals to engage in interactive activities and movements via visual, auditory, and haptic feedback. VR rehabilitation, with elements of imagination, immersion, and interaction, is a novel rehabilitation approach that can improve engagement, motivation, and outcomes for rehabilitation patients [7]. In the early days, VR was applied to stroke rehabilitation. As VR technology has developed and doctors' understanding of the disease has deepened, VR has gradually been applied to numerous rehabilitation fields, such as hand rehabilitation [8],

cardiac function rehabilitation [9], brain function rehabilitation [10], burn rehabilitation [11], and cognitive rehabilitation [12].

In recent years, it has also been gradually applied to postoperative rehabilitation of ACLR. Studies investigating the use of VR in postoperative ACLR rehabilitation have shown conflicting results. While some studies, such as the one by Baltaci et al [13], have not demonstrated significant differences between VR and conventional rehabilitation in terms of muscle strength and dynamic balance, others, like the study by Gokeler et al [14], suggest that VR rehabilitation may have a more significant impact on knee biomechanics and motor learning ability, potentially reducing the risk of a second ACL injury.

VR is increasingly being applied in a variety of rehabilitation fields, which shows the feasibility of VR. Despite promising findings, comparative studies evaluating the efficacy of VR rehabilitation versus conventional rehabilitation in anterior cruciate ligament reconstruction (ACLR) are limited. This meta-analysis aims to critically assess the rehabilitative benefits of VR balance training post-ACLR through a systematic review of relevant randomized controlled trials. The results could inform clinical practice and advance personalized rehabilitation protocols. We hypothesized that VR balance training would be more beneficial than conventional training for postoperative rehabilitation of ACLR patients.

## Materials and methods

### Methods

This study was conducted in accordance with the Preferred Reporting Items for Systematic Evaluation and Meta-Analysis (PRISMA) statement. The PRISMA checklist is provided in the supplementary material. This study was registered with PROSPERO (registration number: CRD42024520383).

**Search strategy.** Nine databases were independently searched by 2 researchers (C.-D. and N.-M.G.). Including 5 English databases (PubMed, Web of Science, Cochrane Library, MEDLINE, Embase) and 4 Chinese databases (China National Knowledge Infrastructure, Chinese Biomedical Literature, China Science and Technology Journal Database, and Wanfang Digital Periodical database). The search time ranged from the inception of each database until 1 January 2024, with no language restriction. The search formula was determined using a combination of MeSH subject terms and free words. The terms mainly included "Virtual Reality", "Educational Virtual Reality", "Educational Virtual Realities", "Instructional Virtual Reality", "Instructional Virtual Realities", "VR", "Anterior Cruciate Ligament", "Cranial Cruciate Ligament", "ACL", "Anterior Cruciate Ligament Injuries", "ACL Injuries", "Anterior Cruciate Ligament Tear", and "ACL Tears". The specific search strategies for the major databases are presented in the Supplementary Material.

### Inclusion and exclusion criteria

**Study design.** To improve the level of evidence, only the articles were included Randomized control trials (RCTs) of VR balance training for rehabilitation after ACLR. Retrospective cohort studies (RCSs) or prospective cohort studies, abstracts or letters were excluded.

**Participants.** Patients after unilateral knee ACLR were included. The included patients were diagnosed according to the recognized criteria for ACL rupture, such as: anterior drawer test and Lachman test, ACL injury or complete rupture as seen on MRI of the knee, and knee arthroscopy meeting the criteria for ACL injury [15]. There were no restrictions on race, gender, age, disease duration, or surgical approach. All patients included in the study were used autografts.

**Intervention and control.** Six RCTs comparing conventional rehabilitation and VR balance training was included. The intervention for patients in the control group was conventional rehabilitation, such as active or passive activities, standing training, plyometric training, and balance training. The intervention for the patients in the treatment group was combined with VR balance training based on the control group. The training intensity, frequency and duration were not restricted.

**Outcome measures.** Primary outcome was the International Knee Documentation Committee (IKDC) score, which was used to assess knee function [16]. The secondary outcomes included: visual analogue scale (VAS) to assess pain level [17]; Holden grading to assess the ability of walking [18]; Extensor peak torque (EPT) and Flexor peak torque (FPT) to assess peripheral knee strength [19]; knee reaction time to assess knee kinematics and knee reproduction angle difference to assess position sense in knee proprioception [20].

## Literature screening

Original literatures searched from the database were imported into Endnote 21. Two researchers (C.-D. and N.-M.G.) independently conducted the literature screening, which firstly excluded duplicates; and then based on the titles and abstracts, we conducted the initial screening to exclude the abstracts, reviews, meta-analyses, syntheses, guides, and dissertations of master's and doctoral theses; the literature of animal experiments and non-clinical trials; registered but unpublished literature with no access to original data; non-randomized controlled trials; and literature with interventions other than VR balance training; and then read the full text further to screen the qualified literature in strict accordance with the set inclusion and exclusion criteria. Included studies were validated by reading the full text. Any disagreements were resolved through a review of the full-text of the original articles by an additional author (T.-C.G.), followed by a discussion.

## Data extraction

Data extraction was performed by two authors (C.-D. and N.-M.G.). After carefully read the full text of the included text, including: first author, year of publication, male number, mean age, interventions, sample size, design, training methods, frequency of VR training, duration of intervention and outcome indicators. In case of disagreement between two researchers, a third researcher (A.-F.L) was consulted. If necessary, we will contact the authors of the original literature for more detailed information.

## Study risk of bias assessment

The Cochrane Risk of Bias tool was applied for randomized trials. Risk of Bias was rated by two independent researchers (C.-D. and N.-M.G.). After assessing all studies (i.e., outcomes) differences were resolved by discussion. Only data that were relevant for the systematic review were extracted from the individual studies.

## Quality assessment of the body of evidence

The quality of evidence within each study was assessed using the Grading of Recommendations Assessment, Development, and Evaluation (GRADE) approach [21]. The quality of the body of evidence for each primary outcome measure was downgraded by criteria based on five domains (risk of bias, inconsistency, indirectness, imprecision, and publication bias) [21]. The interstudy heterogeneity of the synthesized data was assessed using $I^2$, which can be used even when the number of included studies is small [22].

## Publication bias

A funnel plot was generated to assess publication bias, and Egger's test for funnel plot asymmetry was performed in cases where the number of studies in each outcome measure was >10; if the number was ≤10, we assessed funnel plot asymmetry visually.

## Statistical analysis

Meta-analysis was performed by RevMan 5.4 software, the seven outcome indicators in this study were continuous variables, the mean difference (MD) was used to express their effect sizes. and 95% confidence interval (CI) were taken for all indicators. Heterogeneity was tested by the $I^2$ test The heterogeneity across the studies was analyzed using the $I^2$, the significance of heterogeneity was determined using $I^2 \geq 75\%$ or $P < 0.1$. If $P < 0.1$, $I^2 \geq 75\%$, the heterogeneity was large and analyzed by random effect model; If $P \geq 0.10$, $I^2 < 75\%$, the heterogeneity was small and analyzed by fixed effect model [23]. The robustness of the pooled conclusions was assessed through a sensitivity analysis. Subgroup analysis was also conducted based on country, intervention time, and intervention measures. The publication bias for each investigated outcome was assessed through visual inspections of funnel plots.

## Results and discussion

### Results

**Literature search.** Through extensive search, 4447 potentially eligible studies were initially retrieved. By screening the duplicates, 2496 pieces of literature were obtained, after reading the titles and abstracts of the literature, 2141 pieces of literature were excluded which did not match the research direction. 23 pieces of literature were left over after carefully reading the full text. Finally, 6 pieces of literature were finally included for statistical analysis, and the literature was strictly screened according to the inclusion and exclusion criteria, the specific literature screening process is shown in Fig 1. Five literatures in Chinese [24–28] and one in English [29].

**Characteristics of included studies.** Characteristics of included studies are listed in Table 1. A total of 6 RCTs from 2020–2023 [24–29] involving 464 patients after unilateral ACLR were included (232 patients in the treatment group and 232 patients in the control group). The minimum intervention duration was 8 weeks and the maximum was 12 weeks. Detailed rehabilitation methods are in S1 Table. The outcome indicators included IKDC score, knee reaction time, knee angle reproduction difference, VAS score, Holden grading, Extensor peak torque (EPT) and Flexor peak torque (FPT).

**Assessment of risk of bias.** Results of risk of bias assessment for RCTs are shown in Fig 2. This study assessed the risk of bias according to the Cochrane Collaboration's assessment tool Review Manager 5.4, and the methodological quality of the six studies that met the inclusion criteria was evaluated. Five RCTs [24–28] used randomized number tables, and one study [29] used computerized random assignment to generate random sequences, the random sequence entries were rated as "low risk". One study [29] explicitly stated that it was not blinded to the investigators and subjects, which may have some influence on the results of the trial, so the implementation bias was rated as "high risk", while the implementation bias of the remaining studies was rated as "unclear risk"; Six studies did not specify whether the outcome assessors were blinded, the measurement bias was rated as "unclear risk". None of the six studies had sample dropout or loss to follow-up during the trial, the follow-up bias was rated as "low risk". One study [26] failed to present a complete list of expected outcome metrics in the results, selective reporting of the entries was considered "high risk", while the remaining studies had

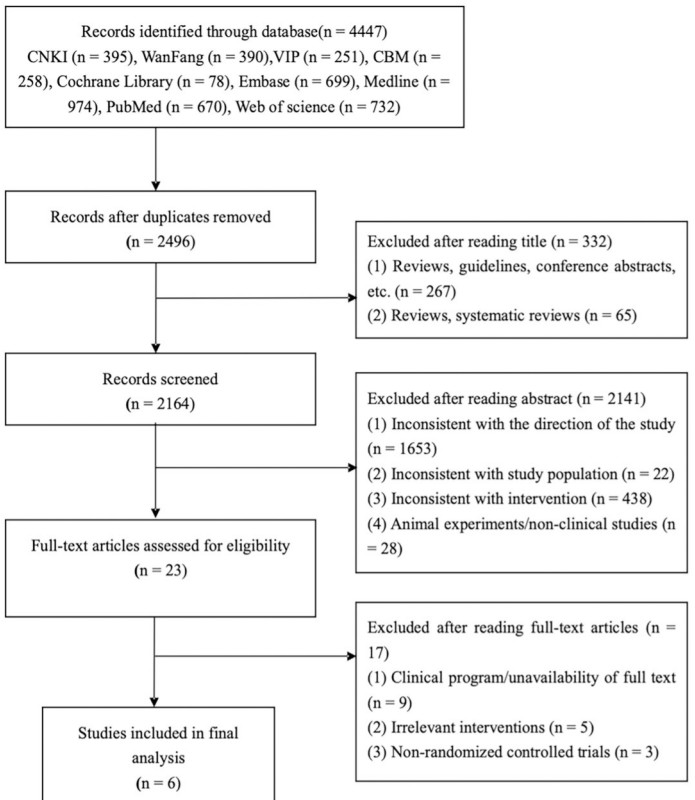

**Fig 1. Flow diagram of the literature search and studies selection process.**

complete outcome data and were rated as "low risk". All studies did not specifically describe allocation concealment and were rated as "unclear risk".

**Quality assessment of the body of evidence.** The results of the GRADE approach are shown in Table 2. The IKDC started with a very low-quality rating. The other outcomes started with a low-quality rating.

**Publication bias.** Since the number of included articles was less than 10, in principle, the funnel plot should not be used to detect publication bias, but we still did this work. The results of funnel plot are shown in the S1 Fig. There is no significant publication bias, but considering the small number of included literatures, this result is not highly credible.

**Primary outcomes.** IKDC score was reported in four RCTs [24, 25, 27, 29]. The results of meta-analysis showed that compared with the control group, VR balance training could significantly improve the IKDC score of patients after ACLR surgery and improve knee joint function (MD = 3.88, 95%CI: 0.95~6.81, $P = 0.01$, $I^2 = 90\%$, Fig 3). Significant heterogeneity was detected ($I^2 = 90\%$, $P = 0.01$) Our sensitivity analysis found the heterogeneity of the conclusion was significantly reduced (MD = 5.30, 95%CI: 3.94~6.65, $P = 0.72$, $I^2 = 0\%$). after excluding 1 study [29] S2 Fig. The result still showed statistical significance. We further explored the IKDC result heterogeneity through subgroup analysis based on country, training frequency, and whether 3D was used in the intervention. The results revealed that, after grouping by country and training frequency, the within—group heterogeneity of each group was not significant. This indicates that the country and training frequency in the literature were among the sources of significant heterogeneity Figs 4 and 5. When grouped by whether 3D was used, the within—

**Table 1. The baseline characteristics of identified studies and patients.**

| First author | Publication year | Male number (%) T | Male number (%) C | Female number (%) T | Female number (%) C | Mean age (Years) T/C | Interventions C | Interventions T | Sample size T/C | Design | Training methods | Frequency of virtual reality training | Duration of intervention | Outcome indicators |
|---|---|---|---|---|---|---|---|---|---|---|---|---|---|---|
| Jin | 2022 | 30(54) | 25(45) | 26(46) | 31(55) | 38.32/40.00 | Routine rehabilitation training | Routine rehabilitation training + Virtual Reality Balance Training | 56/56 | RCT | ITreadmail VR Intelligent Running Table System | 15 min/once, 2 times/d, 5 d/week | 8 weeks | 1,2,4,5,6,7 |
| Li | 2022 | 29(58) | 32(64) | 21(42) | 18(36) | 39.41/40.12 | Routine rehabilitation training | Routine rehabilitation training + Virtual Reality Balance Training | 50/50 | RCT | Dynstable Virtual Balance Trainer | 15–20 min/once, 5 times/week | 8 weeks | 1,3 |
| Shi | 2020 | 26(72) | 28(78) | 10(28) | 8(22) | 31.82/31.44 | Routine rehabilitation training + balance training after 8 weeks | Routine Rehabilitation training + Virtual Reality Balance Training after 7 weeks | 36/36 | RCT | Dynstable VR 3D Balance Training System | 15min/once, 2times/d, 6 d/week | 8 weeks | 2,3 |
| Shi | 2021 | 14(56) | 15(60) | 11(44) | 10(40) | 27.24/27.92 | Routine rehabilitation training | Routine Rehabilitation training + Virtual Reality Balance Training after 5 weeks | 25/25 | RCT | Dynstable VR 3D Balance Training System +Dynstable VR 3D Balance Training System | 15 min/once, 2 times/d, 5 d/week (ITreadmail); 10 min/once, 2 times/d, 5 d/week (Dynstable) | 8 weeks | 1,5,6,7 |
| Zhang | 2021 | unclear | unclear | unclear | unclear | unclear | balance training | Virtual Reality Balance Training | 50/50 | RCT | TechnoBody Virtual Balance Training System Software | 15 min/once, 2 times/d, 6 d/week | 8 weeks | 2 |
| Gsangaya | 2023 | unclear | unclear | unclear | unclear | unclear | Routine rehabilitation training | Routine Rehabilitation training + Virtual Reality Balance Training after 11 weeks | 15/15 | RCT | Dynstable Virtual Balance Trainer | 30 min/once, 1 time/2 weeks | 12 weeks | 1,4 |

RCT: randomized controlled trial; T: treatment group; C: control group; 1: International Knee Documentation Committee score; 2: Knee reaction time; 3: Knee angle reproduction difference; 4: Visual Analogue Scale; 5: Holden grading; 6: Extensor peak torque (EPT); 7: Flexor peak torque (FPT)

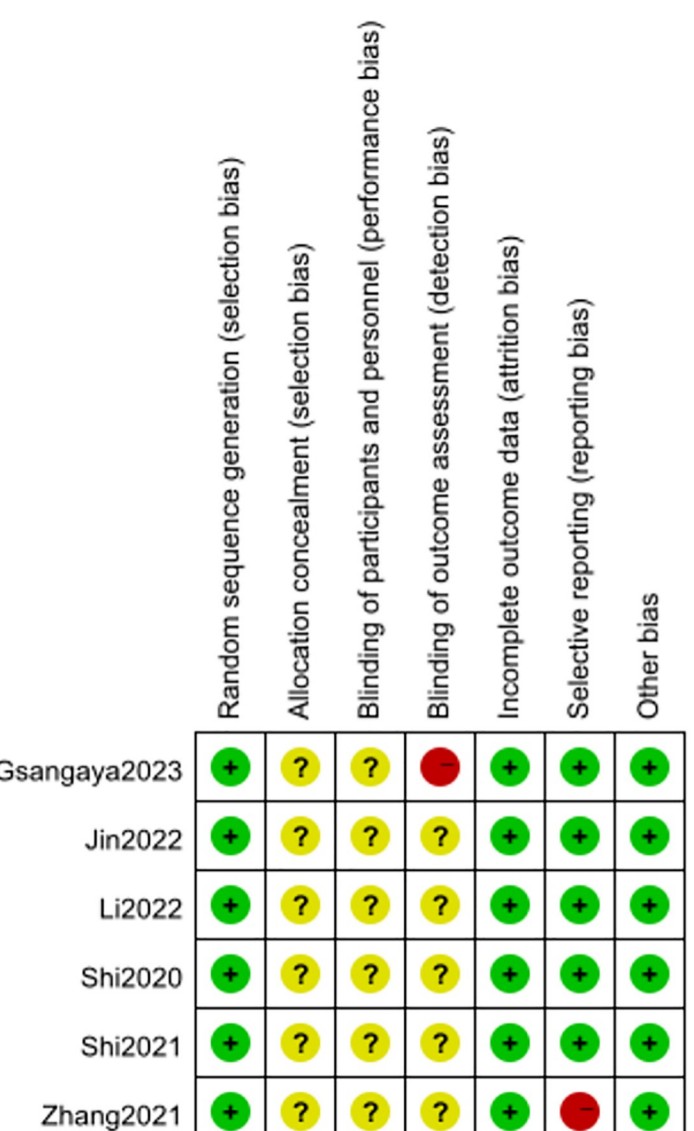

**Fig 2. Risk of bias summary.**

group heterogeneity remained significant in the non—3D group. Interestingly, though, the results in non—3D group were not significantly different (MD = 3.41, 95%CI: -0.004~6.87, $P < 0.0001$, $I^2 = 0\%$). In contrast, the 3D group showed a significant difference (MD = 5.36, 95%CI: 2.78~7.94, $P < 0.0001$). This suggests that 3D may not be the source of IKDC heterogeneity. Summary, the use of 3D may render the IKDC results of VR balance training significant Fig 6.

**Secondary outcomes.** **Knee reaction time** was reported in three RCTs [25–27]. Meta-analysis showed that patients rehabilitated with VR balance training had significantly shorter knee reaction times compared to the control group than patients with conventional rehabilitation (MD = -0.30, 95%CI: -0.35~-0.25, $P < 0.00001$, $I^2 = 0\%$, Fig 7).

Knee angle reproduction difference at 30˚ and 60˚ were reported in two RCTs [24, 28]. The results of the Meta-analysis showed that, compared with the control group, the VR balance

**Table 2. Summary of the evidence according to the GRADE approach.**

| Outcome | Study design | Sample size | Heterogeneity | Level of evidence (GRADE) |
|---|---|---|---|---|
| IKDC | 4 RCT | n = 146 | $I^2 = 90\%$ | ⊕○○○ VERY LOW |
| Knee reaction time | 3 RCT | n = 142 | $I^2 = 0\%$ | ⊕⊕○○ LOW |
| Knee angle reproduction difference | 2 RCT | n = 86 | $30°: I^2 = 16\%$ $60°: I^2 = 53\%$ | ⊕⊕○○ LOW |
| VAS score | 2 RCT | n = 71 | $I^2 = 1\%$ | ⊕⊕○○ LOW |
| Holden grading | 2 RCT | n = 81 | $I^2 = 2\%$ | ⊕⊕○○ LOW |
| EPT | 2 RCT | n = 81 | $I^2 = 91\%$ | ⊕⊕○○ LOW |
| FPT | 2 RCT | n = 81 | $I^2 = 75\%$ | ⊕⊕○○ LOW |

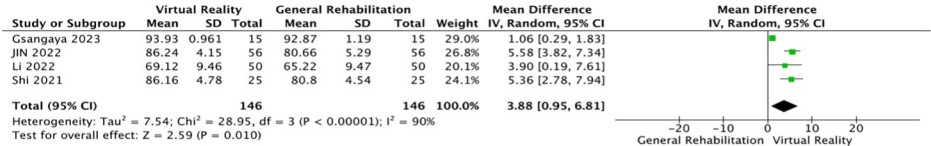

**Fig 3. IKDC score.**

training significantly reduced the knee angle reproduction difference at 30˚ (MD = -0.88, 95% CI: -1.16~-0.61, P<0.00001, $I^2$ = 16%, Fig 7) and 60˚ (MD = -0.80, 95%CI: -1.09~-0.50, P<0.00001, $I^2$ = 53%, Fig 7).

Two RCSs [25, 29] reported the VAS score. Meta-analysis showed that VR balance training significantly reduced VAS pain scores in postoperative patients with ACLR, compared with the control group (MD = -0.52, 95%CI: -0.65~-0.39, P<0.00001, $I^2$ = 0%, Fig 8).

Holden grading was reported in two RCTs [25, 27]. Meta-analysis showed that VR balance training significantly improved Holden grading in patients with ACLR postoperatively compared to the control group (MD = 0.42, 95%CI: 0.33~0.51, P<0.00001, $I^2$ = 0%, Fig 8).

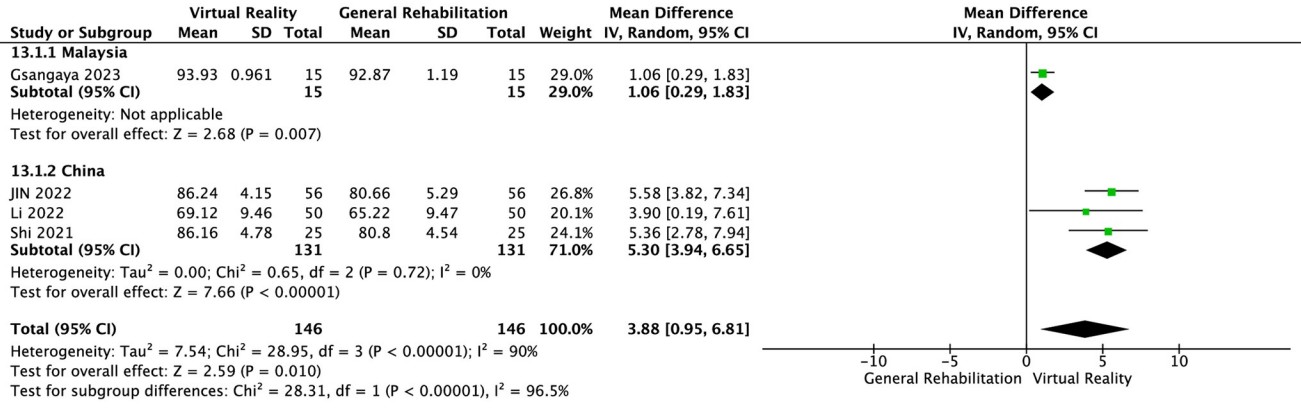

**Fig 4. Subgroup analyses for country (IKDC score).**

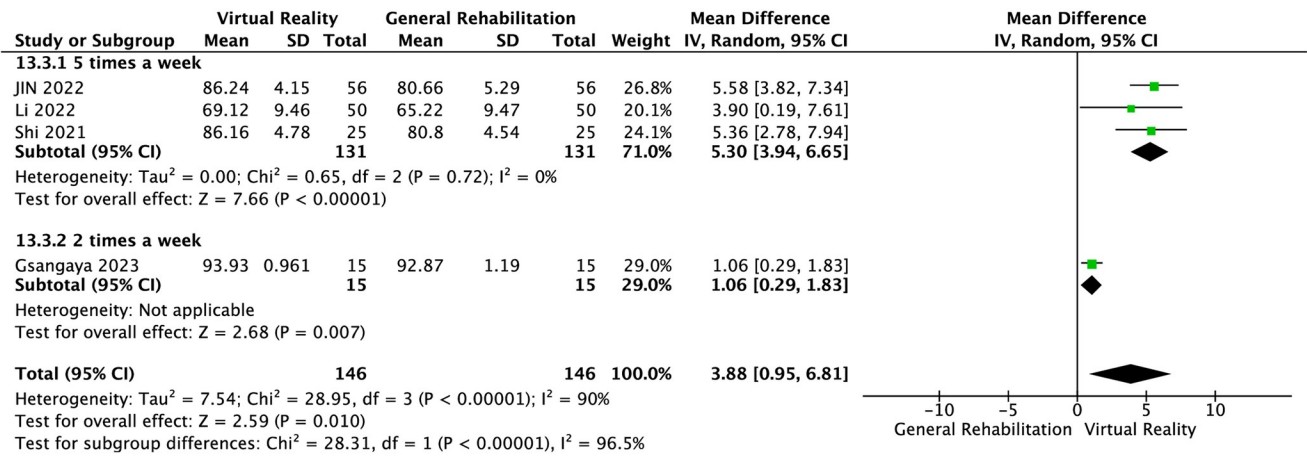

**Fig 5. Subgroup analyses for training frequency (IKDC score).**

EPT was reported in two RCTs [25, 27]. Meta-analysis showed that VR balance training improved the extensor peak torque of postoperative patients with ACLR, compared with the control group (MD = 12.03, 95%CI: 3.28~20.78, $P = 0.007$, $I^2 = 91\%$, Fig 8). The result is highly heterogeneous. However, since only two studies are included, sensitivity analysis and subgroup analysis cannot be carried out. Therefore, the EPT results should be treated with caution.

FPT was reported in two RCTs [25, 27]. Meta-analysis showed that VR balance training improved the flexor peak torque of postoperative patients with ACLR, compared with the control group (MD = 14.57, 95%CI: 9.52~19.63, $P < 0.00001$, $I^2 = 75\%$, Fig 8).

## Discussion

This study examined evidence from previous published studies to investigate the effects of virtual—reality balance training on the rehabilitation following anterior cruciate ligament reconstruction. A quantitative analysis was carried out on data from 464 individuals in six RCTs. Compared with the control group, VR balance training had significant advantages in patients with ACLR in terms of postoperative IKDC scores, knee reaction time, angle reproduction

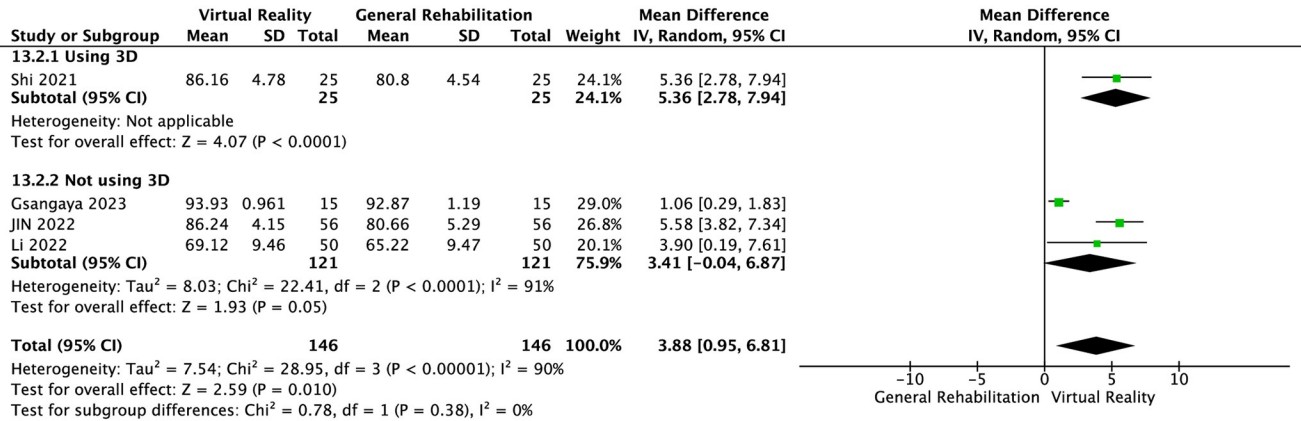

**Fig 6. Subgroup analyses for 3D (IKDC score).**

### A Knee reaction time

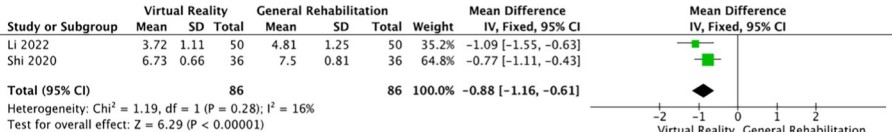

### B Knee angle reproduction difference at 30°

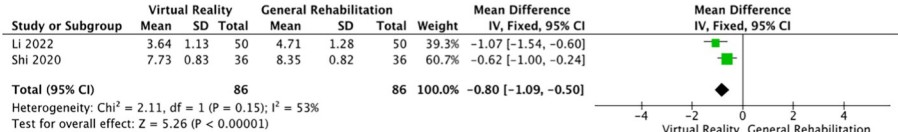

### C Knee angle reproduction difference at 60°

**Fig 7. Secondary outcomes.**

### A VAS score

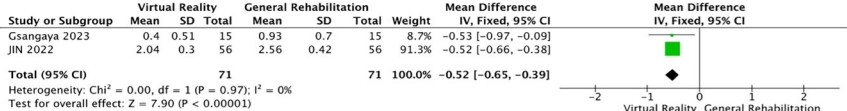

### B Holden grading

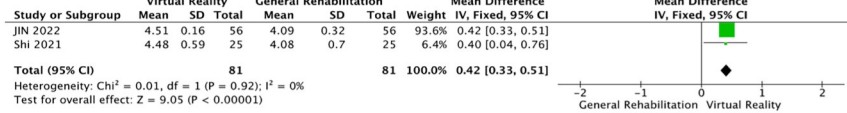

### C Extensor peak torque (EPT)

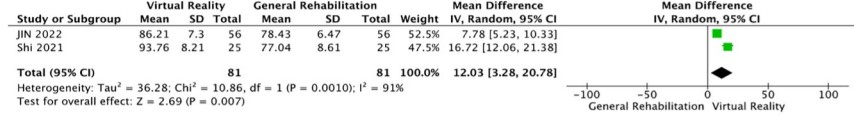

### D Flexor peak torque (FPT)

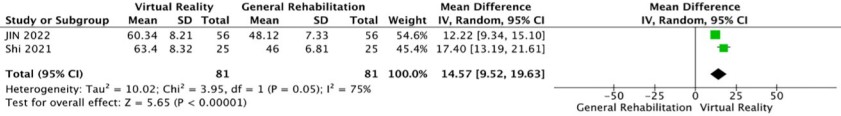

**Fig 8. Secondary outcomes.**

differences at 30˚ and 60˚ of the knees, VAS scores, Holden grading, peak knee extension moment, and peak flexion moment. Additionally, the IKDC score may be affected by country and training frequency. The use of 3D might be beneficial for rehabilitation efficacy.

At present, VR technology has been extensively applied in numerous medical fields, including medical education [30], surgical simulation [31], pain management [32], psychotherapy [12, 33], and telemedicine [34], among others. VR includes desktop VR, immersive VR, augmented VR, distributed VR, mixed reality, and extended reality, etc. In this article, the included studies utilized desktop VR or immersive VR to enhance individuals' balance and body control abilities by simulating diverse balance challenges. This training method can offer an interactive and self—adaptive environment, which is suitable for various groups of people. VR balance exercises come in multiple modes. Immersive VR mainly employs visual feedback and guidance, multi—directional dynamic standing, task—orientation, and balance reaction for balance training. In contrast, desktop—type VR mostly trains balance reaction and coordinated control via lower—limb movements.

Currently, patients still encounter numerous problems following ACLR, such as knee pain [35], knee stiffness, osteoarthritis [36], impairment of knee joint function [37] and proprioception [38], atrophy of lower—limb muscles, and a decline in strength [39]. The results of previous studies have shown that VR combined with rehabilitation robotics training can effectively improve the Holden Walking Function Grade in ACLR patients compared with conventional rehabilitation [40]. This is consistent with the results of this study. Most patients have a slow gait after knee surgery as the body adjusts to increase gait stability and reduce the risk of falls. One study [41] showed that VR training significantly increased gait speed and reduced limitations in activities of daily living in patients with multiple sclerosis. Two other studies [42, 43] showed that VR balance training of stroke patients found a significant increase in gait speed, which supports our findings. Angela Hibbs's research [44] indicates that as the level of immersion rises, the pleasure of exercise also grows. The article by Mark Ehioghae [45] also indicated that immersive VR has a positive influence on proprioception, balance, and pain management in patients following knee—joint surgery. Another systematic review [46] highlights that the application of immersive VR—assisted training is associated with enhancing the enjoyment of physical activity, decreasing perceived fatigue, and strengthening the willingness to exercise. Our research findings also suggest that VR, after the application of 3D technology, appears to be more advantageous for the recovery of knee—joint function. Reducing knee pain is also one of the main purposes of ACL reconstruction. Pain is influenced by a variety of factors such as psychology and length of postoperative recovery [6], and VR training may have an impact on the patient's psychology through human-computer interaction, which in turn may reduce pain. Also, the exercise process included in it may reduce pain by raising the pain threshold or reducing serum inflammation among other pathways [47]. The results of previous studies showed no significant difference between VR training and conventional rehabilitation in terms of postoperative pain relief [48]. In contrast, the results of a recent small-sample study showed that VR training guided by an expert system was superior to conventional rehabilitation in terms of pain relief [49]. This is consistent with the results of this study. The lack of receptor re-innervation after ACL rupture and ACLR surgery can weaken the proprioception of the knee, which may affect the recovery of knee function [50]. Knee angle reproduction difference is a reliable way of assessing knee position sense [20], which responds to the recovery of knee position sense by comparing the angular difference between passive and active activities at a specific angle. Knee reaction time is evaluated through the Dynstable virtual balance assessment system (the Netherlands), which can reflect the response speed of the nervous system and the coordination ability of muscles. The results of previous studies have shown that the use of balance facilitation training significantly improves the recovery of knee

proprioceptive ability in patients after ACLR [51, 52]. The results of this study showed that the addition of a VR balance training group rehabilitated knee joints at 30˚ and 60˚ reproduction difference, as well as knee joint reaction time, were significantly reduced, and the difference was statistically significant. It has been shown that proprioception of joints transmits basic information to motor control areas of the brain such as balance or vestibular sensation through nerves [53]. At the same time, proprioception plays an important role in inducing and stimulating voluntary and involuntary movements. a pilot study [54] found that the immediate visual feedback provided by VR may facilitate joint realignment, thereby activating the proprioceptive sense responsible for joint position perception. On the other hand, VR balance training produces movements that can increase muscle strength through trunk and lower limb motor control. Increased muscle strength, repetitive physical exercises, and increased physical activity also cause recovery of joint proprioception [55]. Muscle strength around the knee also plays an important role in rehabilitation after ACLR [19]. Studies have shown that the periarticular muscles contribute to the stability of the knee joint [56]. On the other hand, adequate muscle strength is helpful for knee neuromuscular control [57]. At the same time, increased strength is essential for restoring proprioception [56]. The results of previous studies have shown that VR combined with isometric plyometric training significantly enhanced periprosthetic knee muscle strength after ACLR compared to conventional rehabilitation [58]. This is consistent with the results of this study.

There were some limitations in this Meta-analysis. One the quality of the included studies was not high, and most of the studies were in Chinese, another is specific scheme and intervention duration of VR balance training in each study were not completely unified, which may affect the stability of the results. Future research can start with VR combined with other rehabilitation methods and be carried out under the strict guidance of experts to develop the best rehabilitation treatment plan and improve clinical efficacy. The research on the effect of surgical methods [59] on the rehabilitation effect of VR training can also be explored. The influence of whether to retain the ACL stump during operation and the choice of single-strand or double-strand reconstruction on the efficacy of VR training is still unknown.

## Conclusions

Since many of the included results are based on low—or very—low—quality evidence, although the results show a certain trend, the conclusion has great uncertainty. In the rehabilitation training following ACLR and lower—limb balance training, the application of VR might be advantageous for the recovery of patients' knee joint function, lower—limb muscle strength, proprioception, and pain management. The level of immersion may influence the rehabilitation outcome. Because of the limitations in data quality and heterogeneity as well as the small sample size, the strength of the conclusions is weakened. These findings should be verified in further large-scale prospective studies.

## Supporting information

**S1 Table. Detailed rehabilitation methods.**
(TIF)

**S1 Fig. Funnel plot.**
(TIF)

**S2 Fig. IKDC score sensitivity analysis.**
(TIF)

**S1 Data. All studies data.**
(XLSX)

**S2 Data. Meta- analyze data.**
(XLSX)

**S3 Data. Publication bias.**
(XLSX)

**S4 Data. Risk of bias.**
(XLSX)

**S1 Checklist. PRISMA 2020 checklist.**
(DOCX)

**S1 File.**
(DOCX)

## Acknowledgments

First, I would like to sincerely thank my supervisor, Ai-Feng Liu. Throughout the entire research process, my supervisor has given me meticulous guidance, from determining the research direction to revising the details of the thesis. Secondly, I would like to thank Nei-Meng Gu and Tian-Ci Guo. In the exchanges and discussions with them, I have obtained a lot of new ideas and inspiration.

## Author Contributions

**Conceptualization:** Ai-Feng Liu.

**Data curation:** Chao Du, Nei-Meng Gu, Tian-Ci Guo.

**Formal analysis:** Chao Du, Nei-Meng Gu.

**Investigation:** Chao Du, Nei-Meng Gu, Tian-Ci Guo.

**Methodology:** Ai-Feng Liu.

**Writing – original draft:** Chao Du, Nei-Meng Gu.

**Writing – review & editing:** Ai-Feng Liu.

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
