## [Decision Letter · Decision Letter 0]

28 Oct 2024

PONE-D-24-41399Efficacy of virtual reality balance training on rehabilitation outcomes following anterior cruciate ligament reconstruction: A systematic review and meta-analysisPLOS ONE

Dear Dr. Liu,

Thank you for submitting your manuscript to PLOS ONE. After careful consideration, we feel that it has merit but does not fully meet PLOS ONE’s publication criteria as it currently stands. Therefore, we invite you to submit a revised version of the manuscript that addresses the points raised during the review process.

**ACADEMIC EDITOR: **We note that one or more reviewers have recommended that you cite specific previously published works. As always, we recommend that you please review and evaluate the requested works to determine whether they are relevant and should be cited. It is not a requirement to cite these works. We appreciate your attention to this request. 

We look forward to receiving your revised manuscript.

Kind regards,

Khalid Taher Mohammed Al-Hussaini, Ph.D

Academic Editor

PLOS ONE

Journal Requirements: When submitting your revision, we need you to address these additional requirements. 1. Please ensure that your manuscript meets PLOS ONE's style requirements, including those for file naming. The PLOS ONE style templates can be found at https://journals.plos.org/plosone/s/file?id=wjVg/PLOSOne_formatting_sample_main_body.pdf and https://journals.plos.org/plosone/s/file?id=ba62/PLOSOne_formatting_sample_title_authors_affiliations.pdf 2. As required by our policy on Data Availability, please ensure your manuscript or supplementary information includes the following:  A numbered table of all studies identified in the literature search, including those that were excluded from the analyses.   For every excluded study, the table should list the reason(s) for exclusion.   If any of the included studies are unpublished, include a link (URL) to the primary source or detailed information about how the content can be accessed.  A table of all data extracted from the primary research sources for the systematic review and/or meta-analysis. The table must include the following information for each study:  Name of data extractors and date of data extraction  Confirmation that the study was eligible to be included in the review.   All data extracted from each study for the reported systematic review and/or meta-analysis that would be needed to replicate your analyses.  If data or supporting information were obtained from another source (e.g. correspondence with the author of the original research article), please provide the source of data and dates on which the data/information were obtained by your research group.  If applicable for your analysis, a table showing the completed risk of bias and quality/certainty assessments for each study or outcome.  Please ensure this is provided for each domain or parameter assessed. For example, if you used the Cochrane risk-of-bias tool for randomized trials, provide answers to each of the signalling questions for each study. If you used GRADE to assess certainty of evidence, provide judgements about each of the quality of evidence factor. This should be provided for each outcome.   An explanation of how missing data were handled.  This information can be included in the main text, supplementary information, or relevant data repository. Please note that providing these underlying data is a requirement for publication in this journal, and if these data are not provided your manuscript might be rejected. 3. Thank you for stating the following financial disclosure: "The funding support from the National Natural Science Foundation of China (Grant no. 81873316) and the Tianjin Health Commission Jinmen medical talent project (Grant no. TJSJMYXYC-D2-028) for this work is gratefully acknowledged." Please state what role the funders took in the study.  If the funders had no role, please state: ""The funders had no role in study design, data collection and analysis, decision to publish, or preparation of the manuscript."" If this statement is not correct you must amend it as needed. Please include this amended Role of Funder statement in your cover letter; we will change the online submission form on your behalf. 4. We note that your Data Availability Statement is currently as follows: All relevant data are within the manuscript and its Supporting Information files. Please confirm at this time whether or not your submission contains all raw data required to replicate the results of your study. Authors must share the “minimal data set” for their submission. PLOS defines the minimal data set to consist of the data required to replicate all study findings reported in the article, as well as related metadata and methods (https://journals.plos.org/plosone/s/data-availability#loc-minimal-data-set-definition). For example, authors should submit the following data: - The values behind the means, standard deviations and other measures reported;- The values used to build graphs;- The points extracted from images for analysis. Authors do not need to submit their entire data set if only a portion of the data was used in the reported study. If your submission does not contain these data, please either upload them as Supporting Information files or deposit them to a stable, public repository and provide us with the relevant URLs, DOIs, or accession numbers. For a list of recommended repositories, please see https://journals.plos.org/plosone/s/recommended-repositories. If there are ethical or legal restrictions on sharing a de-identified data set, please explain them in detail (e.g., data contain potentially sensitive information, data are owned by a third-party organization, etc.) and who has imposed them (e.g., an ethics committee). Please also provide contact information for a data access committee, ethics committee, or other institutional body to which data requests may be sent. If data are owned by a third party, please indicate how others may request data access. 5. Please include your tables as part of your main manuscript and remove the individual files. Please note that supplementary tables (should remain/ be uploaded) as separate ""supporting information"" files 6. Please include captions for your Supporting Information files at the end of your manuscript, and update any in-text citations to match accordingly. Please see our Supporting Information guidelines for more information: http://journals.plos.org/plosone/s/supporting-information.

Reviewers' comments:

Reviewer's Responses to Questions

**Comments to the Author**

1. Is the manuscript technically sound, and do the data support the conclusions?

Reviewer #1: No

Reviewer #2: Partly

2. Has the statistical analysis been performed appropriately and rigorously? 

Reviewer #1: No

Reviewer #2: Yes

3. Have the authors made all data underlying the findings in their manuscript fully available?

Reviewer #1: Yes

Reviewer #2: No

4. Is the manuscript presented in an intelligible fashion and written in standard English?

Reviewer #1: No

Reviewer #2: Yes

5. Review Comments to the Author

Reviewer #1: The manuscript addresses a significant and current topic in rehabilitation science, specifically the application of VR balance training following anterior cruciate ligament reconstruction. While the research question is valuable, the execution of the review and meta-analysis has several methodological weaknesses and areas for improvement in both the writing and statistical analysis.

Major Comments:

o The manuscript reports high heterogeneity in key outcomes such as the International Knee Documentation Committee (IKDC) score (I² = 90%). Although a sensitivity analysis was conducted, the sources of this heterogeneity are not adequately discussed. This raises concerns about the validity of the pooled results. High heterogeneity weakens the reliability of the findings and suggests that the included studies might differ significantly in their methods or patient characteristics. Further subgroup analyses or exploration of potential sources of heterogeneity (e.g., differences in rehabilitation duration or virtual reality training intensity) would strengthen the paper. Please provide a more thorough exploration of potential sources of variability (e.g., differences in intervention duration, VR systems used, patient characteristics). A deeper discussion of how the heterogeneity impacts the reliability of the meta-analysis results is required.

o The GRADE approach reveals that many of the included outcomes are based on low or very low-quality evidence. The manuscript should emphasize the limitations this places on the conclusions. While the authors acknowledge the need for more high-quality studies, the current conclusions seem stronger than what the data supports.

o Consider revising the tone of the conclusions to reflect the low certainty of evidence more clearly. Thus, while the study attempts to draw conclusions about the benefits of virtual reality balance training after ACL reconstruction, the limitations in data quality, heterogeneity, and small sample size weaken the strength of the conclusions.

Minor Comments:

o The manuscript contains several grammatical errors and awkward phrasing. A professional language review is necessary.

o It is not clear what the authors consider as VR. Please elaborate on your definition of VR. For example, do you consider exergames as VR?

o The details about the VR balance training interventions across studies are somewhat vague. Were the VR systems and training protocols comparable? More discussion on the variations in VR systems used (e.g., VR hardware, software, immersion level) would improve the interpretation of the findings.

o The presentation of tables and figures is generally clear, but some could benefit from more descriptive legends to help readers interpret the results without needing to refer back to the text.

o Figure 6 is missing a caption

o Given the significant heterogeneity in some outcomes, have you considered conducting subgroup analyses (e.g., based on age, gender, or duration of intervention)?

o How do you account for potential variations in the quality of virtual reality systems used across the studies? Could differences in the systems affect the outcomes?

o The GRADE assessment resulted in "very low" or "low" quality of evidence for most outcomes. How confident are you in the clinical relevance of the findings, and how would you suggest these results be used in practice?

Introduction:

o Better explain how VR is used in ACLR rehabilitation. As you talk about the potential of VR to "revolutionize ACLR rehabilitation" please make clear how exactly VR support the rehabilitation process.

o You intriduced the abbreviation VR in the introduction but cnstantly write virtual reality. keep it consistent.

o “Despite ACLR being a common procedure, the rate of successful return to pre-injury functional status is not as high as desired, with only around 24% of patients achieving this outcome.” – The phrase "not as high as desired" is vague and not appropriate for a research paper.

Methods

o "The intervention for the patients in the treatment group was combined with virtual reality balance training based on the control group". This is not clear.

o More detailed descriptions of the virtual reality interventions used in the included studies would be helpful. For example, what specific types of virtual reality balance training were applied (e.g., immersive vs. non-immersive VR)

o What is knee reaction time? There is no reference for this parameter

o What is routine rehabilitation (i.e. in table 1)? Be precise in describing the intervention for the CG.

o The manuscript switches between fixed and random effects models depending on the level of heterogeneity, but the rationale for these choices is not fully explained. The choice to use fixed effects for outcomes with lower heterogeneity is valid, but more justification is needed for how these models were selected and how they influence the interpretation of the data.

o Although the manuscript correctly states that the small number of included studies limits the utility of the funnel plot and Egger's test for detecting publication bias, it still includes these analyses.

Discussion

o “Proprioception is transmitted by nerves and plays an important role in joint stability, posture, and motor control.” This sentence does not make sense.

o "In a study [38] in which virtual reality balance training improved peripheral nerve damage, it was found that the improvement in proprioception was most likely achieved due to visual activation in virtual reality." Your reference [38] does not state anything about improved nerve damage following VR balance training nor about improved proprioception. In addition, this study did not use VR, but simple visual biofeedback of ankle motion on a screen. Your interpretation of the findings from [38] are a very far fetch.

o “The exercise process included in it may reduce pain by raising the pain threshold or increasing endogenous paroxysmal substances, among other pathways.” – The term “endogenous paroxysmal substances” is unclear and is completely out of context.

o Location where items are reported in the text are not correct in the Prisma Checklist.

Language

While the manuscript is generally intelligible, there are several grammatical issues and awkward phrasing that affect the clarity of the writing. Some examples include:

a) "The objective of this systematic review and meta-analysis is to clarify the effect the rehabilitation efficacy..." should be rephrased as "The objective of this systematic review and meta-analysis is to clarify the rehabilitation efficacy..."

b) "The prevalence of ACL injuries is relatively high among the younger population..." must be made more precise and concise.

c) “The IKDC score metrics were more heterogeneous, and a sensitivity analysis was performed on them, which revealed that the heterogeneity of the IKDC metrics was significantly reduced.” – The repetition of “IKDC metrics” and “heterogeneity” makes this sentence cumbersome.

d) “At the same time, due to the special physiological structure of women, the results may be different, so the gender should be distinguished in the design of the program and the differences in rehabilitation effects between different genders should be explored.” This sentence is cumbersome and unclear.

e) “[…] virtual reality balance training group rehabilitated a significant reduction in knee VAS scores, and the difference was statistically significant.”

f) “Controversial results were addressed by consulting with a third author when disagreements were encountered, they were discussed in consultation with a 3rd researcher (T.-C.G.).

These are just a few of the many grammatical and stylistic problems impair the readability and clarity of the text. To improve comprehensibility, a thorough language revision is required.

Reviewer #2: Title: report main findings in the title

abstract

any hypothesis?

what key words were used ?

conclusions: coehrent

introduction

too long and confusing

focus on your topic

report the necessity for this systematic review

report the rationale for your study

finish with aim and hypotheiss

methods

very well reported

results

too long try to reduce

discussion

start with main findings of your paper

report what Is new

analyze current literature

improve limitations

highlight clinical impact of your study

conclusions

coherenet

references

add following

Loucas M, Loucas R, D'Ambrosi R, Hantes ME. Clinical and Radiological Outcomes of Anteromedial Portal Versus Transtibial Technique in ACL Reconstruction: A Systematic Review. Orthop J Sports Med. 2021 Jul 2;9(7):23259671211024591. doi: 10.1177/23259671211024591. PMID: 34277881; PMCID: PMC8255613.

D'Ambrosi R, Valli F, Di Feo F, Marchetti P, Ursino N. Use of tourniquet in anterior cruciate ligament reconstruction: Is it truly necessary? A prospective randomized clinical trial. J Orthop Surg (Hong Kong). 2024 Sep-Dec;32(3):10225536241293538. doi: 10.1177/10225536241293538. PMID: 39418227.

D'Ambrosi R, Carrozzo A, Meena A, Corona K, Yadav AK, Annibaldi A, Kambhampati SBS, Abermann E, Fink C. A slight degree of osteoarthritis appears to be present after anterior cruciate ligament reconstruction compared with contralateral healthy knees at a minimum of 20 years: A systematic review of the literature. J Exp Orthop. 2024 Apr 4;11(2):e12017. doi: 10.1002/jeo2.12017. PMID: 38577065; PMCID: PMC10993150.

6. PLOS authors have the option to publish the peer review history of their article (what does this mean?). If published, this will include your full peer review and any attached files.

Reviewer #1: No

Reviewer #2: No

---

## [Author Response · Author response to Decision Letter 0]

19 Nov 2024

Dear Editors and Reviewers,

Thank you for your letter and for the reviewers’ comments concerning. We have given point-to-point answers to each question. Due to the large amount of content, we will upload it as a material, and name the material "Response to Reviewers".

---

## [Decision Letter · Decision Letter 1]

10 Dec 2024

Efficacy of virtual reality balance training on rehabilitation outcomes following anterior cruciate ligament reconstruction: A systematic review and meta-analysis

PONE-D-24-41399R1

Dear Dr. Liu,

We’re pleased to inform you that your manuscript has been judged scientifically suitable for publication and will be formally accepted for publication once it meets all outstanding technical requirements.

Kind regards,

Khalid Taher Mohammed Al-Hussaini, Ph.D

Academic Editor

PLOS ONE

Additional Reviewers' comments:

Reviewer's Responses to Questions

**Comments to the Author**

1. If the authors have adequately addressed your comments raised in a previous round of review and you feel that this manuscript is now acceptable for publication, you may indicate that here to bypass the “Comments to the Author” section, enter your conflict of interest statement in the “Confidential to Editor” section, and submit your "Accept" recommendation.

Reviewer #2: All comments have been addressed

2. Is the manuscript technically sound, and do the data support the conclusions?

Reviewer #2: Yes

3. Has the statistical analysis been performed appropriately and rigorously? 

Reviewer #2: Yes

4. Have the authors made all data underlying the findings in their manuscript fully available?

Reviewer #2: Yes

5. Is the manuscript presented in an intelligible fashion and written in standard English?

Reviewer #2: (No Response)

6. Review Comments to the Author

Reviewer #2: the article deserve to be considered for publication after all the revisions that have been performed by the authors

7. PLOS authors have the option to publish the peer review history of their article (what does this mean?). If published, this will include your full peer review and any attached files.

Reviewer #2: No

---

## [Editor Report · Acceptance letter]

23 Dec 2024

PONE-D-24-41399R1 

PLOS ONE

Dear Dr. Liu, 

I'm pleased to inform you that your manuscript has been deemed suitable for publication in PLOS ONE. Congratulations! Your manuscript is now being handed over to our production team.

Kind regards, 

on behalf of

Assoc. Prof. Dr Khalid Taher Mohammed Al-Hussaini 

Academic Editor

PLOS ONE